# Couples data from north-western Tanzania: Insights from a survey of male partners of women enrolled in the MAISHA cluster randomized trial of an intimate partner violence prevention intervention

Tanya Abramsky[1]*, Imma Kapinga[2], Gerry Mshana[2,3], Shelley Lees[1], Christian Holm Hansen[4], Ramadhan Hashim[2], Heidi Stöckl[1], Saidi Kapiga[2,5‡], Sheila Harvey[1,2‡]

1 Department of Global Health and Development, London School of Hygiene and Tropical Medicine, London, United Kingdom, 2 Mwanza Intervention Trials Unit, Mwanza, Tanzania, 3 National Institute for Medical Research, Mwanza, Tanzania, 4 MRC Tropical Epidemiology Group, London School of Hygiene and Tropical Medicine, London, United Kingdom, 5 Department of Infectious Diseases Epidemiology, London School of Hygiene and Tropical Medicine, London, United Kingdom

‡ These authors are joint senior authors on this work.
* Tanya.abramsky@lshtm.ac.uk

**Data Availability Statement:** Data are available, with restricted access, via the London School of

## Abstract

### Introduction

Globally, around 30% of ever-partnered women have experienced physical and/or sexual intimate partner violence (IPV) during their lifetime. To date, most research into causes and prevention of IPV involves surveys of women, with little research into risk-factors for male perpetration. This paper describes a survey of male partners of women participating in the MAISHA cluster randomised trial (CRT) of an IPV prevention intervention, in Mwanza City, Tanzania. Using linked couples' data, we explore man-, woman-, and relationship-/household-level factors associated with physical and sexual IPV.

### Methods

Women were interviewed at baseline and 29-months follow-up. At follow-up, women were asked for consent to invite their partner to participate in the male survey. We describe response rates for the women's follow-up and male partners' surveys, and identify factors associated with women's consent to approach partners. Multivariate logistic regression was used to explore factors associated with women's past-year experiences of physical and sexual IPV.

### Results

512 (65%) partnered women consented for the partner to be approached, higher among intervention than control women. 425 (83%) male partners were interviewed. Women consenting were disproportionately likely to be in longer-term relationships. Past-year IPV was

Hygiene and Tropical Medicine data archive, Data Compass (DOI: https://doi.org/10.17037/DATA. 00001773). Requests for data access should be submitted via this repository, with applicants outlining the analysis they wish to perform and the data variables requested. Queries and data requests may also be submitted to the LSHTM Research Data Management Service: researchdatamanagement@lshtm.ac.uk (please include the above dataset DOI in all correspondence). Access to an anonymised subset of data will be provided for use in ethically approved research, on condition those requesting access comply with ethical conditions associated with the study. If the proposed use is approved, applicants will be asked to sign a Data Transfer Agreement that requires them to keep data confidential.

**Funding:** The MAISHA study was funded by an anonymous donor and supported by the STRIVE Research Consortium, which is funded by UK Aid (Grant number PO5244 held by SL) from the UK government Department for International Development (https://www.ukaiddirect.org/). The views expressed in the paper do not necessarily reflect the Department's official policies. The funders had no role in study design, data collection and analysis, decision to publish, or preparation of the manuscript.

**Competing interests:** The authors have declared that no competing interests exist.

associated with lower consent among control women, but greater consent in the intervention arm. Factors associated with increased odds of physical IPV were women's childhood experiences of abuse, young age, women's lower income and women's attitudes justifying IPV. Sexual IPV was associated with women's childhood experiences of abuse, young age, educational disparity within couple, men's alcohol use and women's poor mental health.

## Conclusions

We successfully conducted a survey of male partners with the full consent of women trial participants. The breadth of factors associated with IPV demonstrates the need for IPV prevention interventions to work with women and men, and specifically couples. Interventions should address risk-factors for both physical and sexual IPV.

## Introduction

Violence against women (VAW) is a global public health problem that has far reaching consequences for women's mental and physical health, including increased risk of HIV infection [1–3]. The most common perpetrators of VAW are male intimate partners [4], with 30% of ever-partnered women estimated to have experienced physical and/or sexual IPV at some point in their lives [5]. Alongside recognition of the complex interplay of factors (individual-woman-, individual-man-, relationship-, community- and societal-level), that influence IPV risk, VAW programming has expanded from a focus on responses to survivors and perpetrators, to primary prevention aimed at reducing violence by addressing the underlying risk factors that drive it [6].

Research into the causes of IPV and the mechanisms through which prevention interventions may work, is often limited by lack of data from multiple levels of the social ecology. Understandably, much of the research into VAW has involved surveys of women, with a host of risk factors such as young age, low education, childhood experiences of abuse, harmful alcohol use, attitudes accepting of violence, and educational disparity between partners shown to be associated with women's increased risk of experiencing IPV [6, 7]. Until recently however, there has been little research into risk factors for male perpetration of IPV [8], particularly in low income countries [9].

While surveys of women often include questions on partner characteristics, these are usually restricted to factual questions on demographics and visible behaviours (such as alcohol use), which she could reasonably (though not always reliably) be expected to know. Thus, data are lacking on important partner characteristics such as his attitudes, sexual behaviour, mental health and childhood experiences of violence. On the other hand, there are concerns that men, while providing more accurate information about their own characteristics, may underreport perpetration of IPV [10–13], especially within the context of intervention research where social desirability bias may be exacerbated. Though recent notable studies have successfully addressed the methodological challenges in collecting IPV perpetration data from men [8, 14], these are few and far between and have tended to use measures of lifetime (rather than past year) perpetration of IPV. Identified risk factors for perpetration include childhood experiences of abuse, permissive attitudes towards VAW, inequitable gender attitudes, a feeling of not having attained the 'husband ideal', having multiple sexual partners, involvement in fights (non-partner), alcohol misuse, depression, low education and poverty [8, 14–16].

There is an even greater paucity of VAW-specific studies that have interviewed both members of the same couple ('dyad') [17–19]. While DHS surveys have provided couples data that have been used for multi-country VAW analyses [20], the questions covered in the DHS male survey are limited with regards to important risk factors for violence, such as childhood experiences of violence. Significant safety concerns surround VAW-related dyad research, as women may be put at risk if their partners are made aware of the nature of questions that the women have been asked and the possible disclosures they may have made. Indeed, World Health Organization (WHO) recommendations on researching violence against women have long advised that only one woman per household is interviewed about her experiences of partner violence so as to avoid such suspicion on the part of other household members [21]. However, as the field of violence prevention moves towards interventions that work with both women and men to prevent VAW [22], the value of 'dyad' data is increasingly recognised, including in the newer WHO "Ethical and safety recommendations for intervention research on violence against women" [23]. Not only are couple's data sometimes necessary to evaluate prevention interventions—they also facilitate exploration of the full gamut of factors (relating to both male and female partners) that are associated with women's experiences of IPV, and the full scope of ways in which VAW interventions may impact on relationships.

In this paper, we describe the process and selection biases associated with conducting a survey of male partners of women participating in the MAISHA cluster randomised trial (CRT) of a social empowerment intervention to prevent IPV, in Mwanza City, Tanzania. Using the resulting linked couples' data, we explore how factors operating at different levels of the social ecology (man, woman, and relationship-/ household-level) are associated with women's past year experience of physical and sexual IPV in this setting.

## Methods

The study utilized linked data from a CRT of women in Mwanza city, Tanzania, and a cross-sectional survey of their male partners. The CRT (described in detail elsewhere) [24] involved 66 established microfinance loan groups (comprising a total of 1049 women), randomised to either receive the MAISHA intervention at the outset or be wait-listed for the intervention (control). The intervention comprised a 10-session participatory curriculum that aimed to empower women, prevent IPV and promote healthier relationships. Individual-level written informed consent was obtained from all women before they could participate in the study, though refusal to participate had no bearing on their continued involvement in microfinance. Enrolled women were interviewed face-to-face at baseline and again 29 months later (follow-up), using a structured questionnaire (S1 and S2 Questionnaires). Interviews were conducted in private by female interviewers trained in interviewing techniques, gender issues, violence and ethical issues related to research on IPV [25]. Responses were entered directly onto electronic tablets programmed to check for accuracy and consistency of data entered. The intervention was associated with a reduction in past year experience of physical IPV at follow-up (adjusted odds ratio of physical IPV in intervention women compared to control women: aOR = 0.64, 95%CI 0.41–0.99) [26].

During follow-up, MAISHA study staff met with women at their weekly microfinance loan group meetings and provided them with information (and the opportunity to ask questions) about the male cross-sectional survey. They explained that the survey was linked to the MAISHA study with the aim of exploring men's views on healthy relationships. At the end of each woman's one-to-one follow-up interview, the interviewer described the purpose of the male survey again, providing her with additional opportunities to ask questions. She was then asked whether she would permit a team member to approach her partner to take part. The

interviewer emphasised that they would not contact her partner if they did not have her permission to do so, and that she could withdraw permission if she subsequently changed her mind (her consent was re-confirmed before the partner was actually contacted). Where permission was granted, the woman was consulted on how best to approach her partner–for example whether she would prefer to talk to her partner first before arranging for a member of the team to meet with him. Where interviewers encountered difficulties contacting a partner or scheduling an interview with him, they re-contacted the woman to check whether circumstances had changed and whether it was still okay to persist. This process was designed to prevent women facing excess risk as a result of participating in the research.

Interviews with male partners were conducted by male interviewers, face-to-face, using a structured questionnaire (S3 and S4 Questionnaires). Interviewer training and conduct followed the same procedures as for the women's interviews. All men who were approached were provided with information about the study, and written consent was obtained prior to proceeding with the interview. The study followed WHO recommendations on research on VAW [21]. As with the women's interviews, the study was introduced in broad terms, as a study on health and well-being. The male survey included sections on demographics, childhood experiences, health, attitudes, and relationship characteristics and dynamics. Men were not asked about perpetration of violence against their partners, as inclusion of such questions could alert men to disclosures their partner may have made against them, thereby putting women at risk [23]. Questions on attitudes towards IPV were embedded in a section on broader norms and attitudes (including equitable as well as conservative norms), and introduced in neutral terms, so as not to cast men as the aggressors within relationships. In the same vein, questions on relationship dynamics covered positive as well as negative interactions within the couple.

Any male or female respondents who reported experiencing violence or other difficulties within their relationships were given help to access appropriate services and support. Information about local support services was provided to all participants irrespective of whether or not they reported experiencing violence [24].

The trial and the cross-sectional survey of male partners were approved by the Tanzanian National Health Research Ethics Committee of the Tanzania National Institute for Medical Research (Ref: NIMR/HQ/R.8a/Vol.IX/1512), and the research ethics committee of the London School of Hygiene and Tropical Medicine (LSHTM, Ref: 11642). The research was implemented by the Mwanza Intervention Trials Unit (MITU), the Tanzania National Institute for Medical Research (NIMR) and LSHTM in close collaboration with local leaders and members of the participating communities.

The CRT is registered with ClinicalTrials.gov, number NCT02592252. (https://clinicaltrials.gov/ct2/show/NCT02592252)

## IPV outcomes

The details of variables analysed for this paper are included in S1 Table, and briefly described in the statistical analysis section below. The analysis of factors associated with IPV considered two outcomes measured at follow-up, the woman's past year experience of physical IPV and sexual IPV, reported by the woman as having been perpetrated by her current partner (the one interviewed). Women who reported past year violence perpetrated by a previous partner only were counted as not having the outcome. Violence questions were based on those used in the WHO Multi-country Study on Health and Domestic Violence [4], and asked women about their experience of specific violent acts by a partner. Those with past year experience of any of the physical acts were coded as having past year physical IPV, and any of the sexual acts as past year sexual IPV (see S1 Table for more detail).

## Exposure variables for IPV analysis

The exposures considered in the analysis of factors associated with IPV relate to several layers of the social ecology: the woman (her childhood experiences, demographics, attitudes and behaviours); her male partner (his childhood experiences, demographics, attitudes and behaviours); and the relationship/household (factors pertaining to the relationship itself as well as to the broader household in which the couple lives). We drew up a conceptual framework outlining the hierarchical nature in which these factors relate to the woman's risk of past year IPV experience (Fig 1). The framework separates out factors relating to (1) the childhood of the woman and her partner, (2) current demographics pertaining to the woman, her partner and the relationship/household; and (3) the attitudes/behaviours/health of the woman and her partner.

## Statistical analysis

The analysis utilizes data from the follow-up interviews with women, and the cross-sectional survey of their male partners. Data analysis was done using STATA (version 16) [27].

We compared response rates to the male survey (including whether or not women consented for their partner to be approached) between the intervention and control arm, using cross tabulation.

To identify other factors associated with women's willingness for their partner to be approached, we tabulated women's characteristics (demographics, experiences of abuse, mental health, attitudes to IPV) and relationship characteristics/dynamics (marital status, number of wives, her reports on quality of communication with partner, and how often she had seen him drunk) according to their partner's participation status in the male survey (she did not grant permission to approach him; she granted permission but he refused/could not be located; he completed the survey). Data were first disaggregated by trial arm to check for

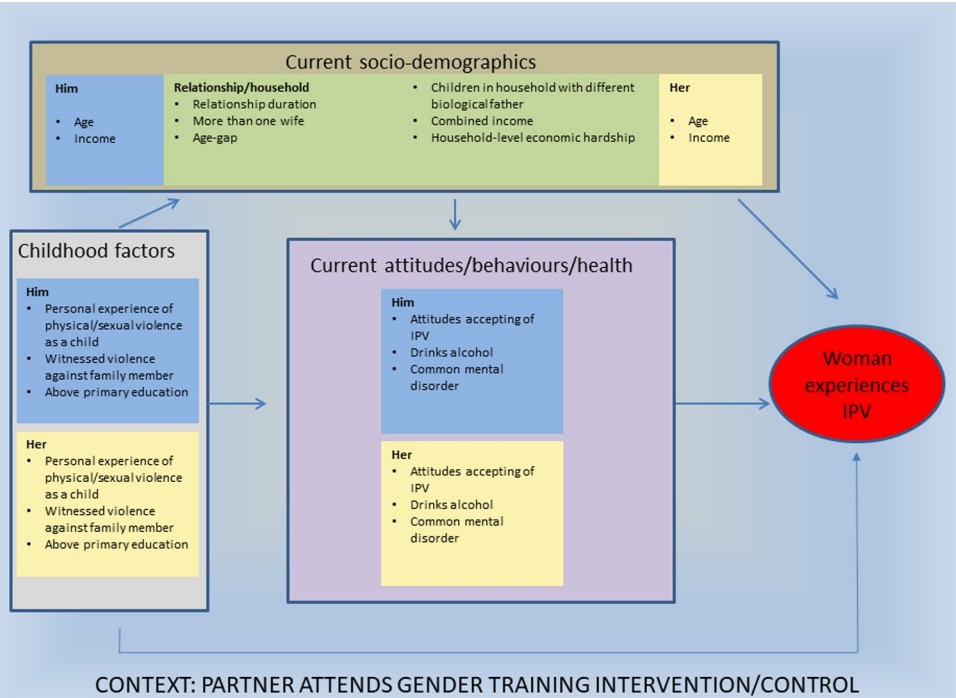

**Fig 1. Conceptual framework of hierarchical relationships between factors associated with past year IPV.**

potential differences in associated factors between women in intervention and control arms. Where differences were observed, we present the data disaggregated by trial arm–otherwise we present data pooled across the two arms. As a supplementary analysis, we performed crude logistic regression to estimate odds ratios of association between the above factors and women's consent for their partner to be approached, and used likelihood ratio tests to assess whether the associations differed between the trial arms.

Data on male participant's socio-demographic characteristics, alcohol use, mental health, gender attitudes, and childhood experiences of IPV/abuse are presented disaggregated by trial arm.

The analysis of factors associated with IPV was restricted to cohabiting/married couples for whom both the man's and woman's data was available. Crude associations between these characteristics/factors and the woman's past year risk of IPV by this partner were explored using cross-tabulations and logistic regression including age as a covariate. Multivariate logistic regression was then conducted including only those variables which were associated (Wald test $p < 0.1$) with either of the IPV outcomes in the simple age-adjusted logistic regression model. In line with our conceptual framework, blocks of variables were added to the model sequentially so that we could estimate the association between more distal exposures and IPV without attenuation by variables potentially on the causal pathway between the two. Variables were added as follows: childhood factors [model 1]; + current socio-demographic characteristics [model 2]; + current attitudes, behaviours and health [model 3]. Cluster robust standard errors (with loan group as the cluster variable) were used to take account of the clustered nature of the data. The analysis was initially performed separately for intervention and control arms, since intervention exposure could modify which factors are associated with IPV. As no substantial differences in associated factors were observed between arms, a pooled analysis is presented.

Where two factors were highly correlated, only one was selected for inclusion in the multivariate analysis, or where conceptually appropriate a composite indicator derived from both was used.

In recognition of low study power in relation to some of the associations under study (the CRT sample size having been determined around power to detect intervention impacts on primary outcomes), interpretation of results will involve consideration of the magnitude and direction of observed associations in addition to p-values and 95% confidence intervals.

## Results

### Factors associated with women's willingness for their partner to be interviewed

Response rates to the male survey, disaggregated by trial arm are presented in Table 1. Of the 1049 women enrolled into the MAISHA CRT, 938 (89%) were interviewed at follow-up, of

**Table 1. Response rates for MAISHA follow-up survey and male partners survey, disaggregated by trial arm.**

|  | Control | Intervention | Total |
|---|---|---|---|
|  | n/N (%) | n/N (%) | n/N (%) |
| Interviewed at trial follow-up | 442/505 (88%) | 496/544 (91%) | 938/1049 (89%) |
| Has a current partner (among those interviewed at follow-up) | 375/442 (85%) | 415/496 (84%) | 790/938 (84%) |
| Consent to approach partners (among those with current partner) | 230/375 (61%) | 282/415 (68%) | 512/790 (65%) |
| Partner interviewed (among those consenting for partner to be approached) | 180/230 (78%) | 245/282 (87%) | 425/512 (83%) |

whom 790 (84%) reported having had a partner in the past year. Of these, 512 (65%) consented for their partner to be invited to participate in the male survey, and 425 male partners (83%) went on to complete an interview. Consent rates were higher among women in the intervention compared to control arm (68% versus 61%), as were completion rates among men approached to participate (87% versus 78%).

Several important differences were seen between women who refused to their partner being approached, and women who consented. These differences were observed across both intervention and control arms of the trial. Women who consented were more likely to be married/living as married than women who refused (93% of women whose male partner was interviewed, versus 62% of women who refused), and to be with the same partner they were with at the time of the trial baseline interview (97% where man interviewed versus 84% where she refused). They were less likely to report that their partner had more than one wife (21% where man interviewed versus 41% where she refused). Those consenting reported better communication with their partner (discussing their day/feelings; being asked advice by their partner) and greater confidence to assert an opinion different to that of their partner. They were also less likely to contribute more financially to the household than their partner (Table 2).

Differences between women in the three response categories were also seen with respect to past year experiences of violence by their current male partner (Table 3). However, the nature of these differences varied between intervention and control arms. In the control arm, women who consented were less likely to have experienced past-year physical and/or sexual IPV than women who refused. The reverse was seen in the intervention arm, where women who consented were more likely than those who refused, to have experienced past-year physical/sexual IPV. The supplementary logistic regression analysis confirmed a statistically significant difference in patterns of association between the two trial arms (see S2 Table). The relative odds of consent among women with past year physical/sexual IPV compared to those without past year physical/sexual IPV was 0.68 (95%CI 0.47–0.97) in the control arm, and 1.88 (95%CI 1.10–3.20) in the intervention arm, with the likelihood ratio test providing strong evidence in support of an interaction between trial arm and IPV (p = 0.003). The same patterns were observed when physical and sexual IPV were examined separately

## Characteristics of men participating in the male partner survey

Men taking part in the male survey were somewhat older than their female partners (mean of 48.0 years for men, versus 40.8 years for their female partners). Men in the intervention arm were on average slightly younger than men in the control arm (Mean of 46.7 years versus 49.8 years). Almost all participants were married or living as married, with 90% formally married. The vast majority (82%) were in relationships of more than 10 years duration. Almost all had biological children, with half also reporting non-biological children living in their household (Table 4).

Overall, 30% of men had above primary-level education, slightly higher in the control arm compared to intervention arm. More than three-quarters (77%) were self-employed, and a similar number (76%) reported household-level financial hardship in the past year (in contrast to 58% of their female partners).

More than a third (37%) of men screened positive for poor mental health (measured using the SRQ20, threshold of > = 8). A high proportion (62%) reported never drinking alcohol, and a third (33%) reported attitudes accepting of men's use of physical IPV against their wives.

Forty percent of male partners surveyed had been beaten as a child at least a few times, with 14% saying they had been hit so hard that they been marked or injured. Ten percent had themselves experienced sexual abuse as a child.

**Table 2. Characteristics of women according to interview status of male partners.**

| | Woman refused for partner to be approached | Woman consented but male partner not interviewed | Male partner interviewed | Total |
|---|---|---|---|---|
| | N = 278 | N = 87 | N = 425 | N = 790 |
| | n/N (%) | n/N (%) | n/N (%) | n/N (%) |
| Currently married/living as married | 173/278 (62%) | 72/87 (83%) | 397/425 (93%) | 642/790 (81%) |
| Same partner as at baseline | 228/272 (84%) | 76/86 (88%) | 403/415 (97%) | 707/773 (91%) |
| Male partner has other wife | 107/262 (41%) | 14/83 (17%) | 85/406 (21%) | 206/751 (27%) |
| Age | | | | |
| <30yrs | 22/272 (8%) | 6/86 (7%) | 40/415 (10%) | 68/773 (9%) |
| 30-39yrs | 98 (36%) | 39 (45%) | 158 (38%) | 295 (38%) |
| 40-49yrs | 114 (42%) | 33 (38%) | 144 (35%) | 291 (38%) |
| 50+ | 38 (14%) | 8 (9%) | 73 (18%) | 119 (15%) |
| Partner's age | | | | |
| <40yrs | 64/257 (25%) | 24/82 (29%) | 97/411 (24%) | 185/750 (25%) |
| 40-49yrs | 89 (35%) | 30 (37%) | 153 (37%) | 272 (36%) |
| 50+ | 101 (39%) | 26 (32%) | 161 (39%) | 288 (38%) |
| Don't know | 3 (1%) | 2 (2%) | 0 (0%) | 5 (1%) |
| Education | | | | |
| None/incomplete primary | 35/272 (13%) | 8/86 (9%) | 58/415 (14%) | 101/773 (13%) |
| Completed primary | 176 (65%) | 57 (66%) | 271 (65%) | 504 (65%) |
| Attended secondary or higher | 61 (22%) | 21 (24%) | 86 (21%) | 168 (22%) |
| Has at least one child | 255/278 (92%) | 81/87 (93%) | 401/425 (94%) | 737/790 (93%) |
| Monthly income* | | | | |
| 1st quartile (lowest) | 67/278 (24%) | 12/87 (14%) | 111/425 (26%) | 190/790 (24%) |
| 2nd quartile | 63 (23%) | 23 (26%) | 80 (19%) | 166 (21%) |
| 3rd quartile | 60 (22%) | 19 (22%) | 97 (23%) | 176 (22%) |
| 4th quartile (highest) | 63 (23%) | 26 (30%) | 93 (22%) | 182 (23%) |
| Doesn't earn | 14 (5%) | 6 (7%) | 25 (6%) | 45 (6%) |
| Don't know | 11 (4%) | 1 (1%) | 19 (4%) | 31 (4%) |
| Contributes more financially to household than partner does | 127/278 (46%) | 35/87 (40%) | 141/425 (33%) | 303/790 (38%) |
| Currently in receipt of a microfinance loan from BRAC** | 151/278 (54%) | 53/87 (39%) | 254/425 (60%) | 458/790 (58%) |
| Experienced household financial hardship in past year | 156/278 (56%) | 46/87 (53%) | 246/425 (58%) | 448/790 (57%) |
| Poor mental health | 102/278 (37%) | 33/87 (38%) | 173/425 (41%) | 308/790 (39%) |
| Seen partner drunk many times in the past year | 56/275 (20%) | 18/87 (21%) | 70/424 (17%) | 144/786 (18%) |
| Attitudes condoning IPV | 140/278 (50%) | 45/87 (52%) | 204/425 (48%) | 389/790 (49%) |
| Discusses what happens in day/feelings with partner | 126/278 (45%) | 55/87 (63%) | 268/425 (63%) | 449/790 (57%) |
| Very confident to assert an opinion if different to partner's | 182/278 (65%) | 65/87 (75%) | 314/425 (74%) | 561/790 (71%) |

*Women were ranked from poorest to wealthiest, based on reported income, and then grouped into four income quartiles (1st being lowest 25% and 4th being highest 25%).

**All women enrolled in the MAISHA trial were members of a microfinance loan group administered by the international development NGO, BRAC.

**Table 3. Women's experience of IPV according to interview status of male partner, disaggregated by trial arm.**

| | Control | | | | Intervention | | | |
|---|---|---|---|---|---|---|---|---|
| | Woman refused for partner to be approached | Woman consented but male partner not interviewed | Male partner interviewed | Total–control | Woman refused for partner to be approached | Woman consented but male partner not interviewed | Male partner interviewed | Total–intervention |
| | N = 145 | N = 50 | N = 180 | N = 375 | N = 133 | N = 37 | N = 245 | N = 415 |
| | n/N (%) | n/N (%) | n/N (%) | n/N (%) | n/N (%) | n/N (%) | n/N (%) | n/N (%) |
| Past year experience of sexual and/or physical IPV | 54 (37%) | 12 (24%) | 52 (29%) | 118 (31%) | 26 (20%) | 12 (32%) | 76 (31%) | 114 (27%) |
| Past year experience of physical IPV | 39 (27%) | 9 (18%) | 33 (18%) | 81 (22%) | 16 (12%) | 6 (16%) | 47 (19%) | 69 (17%) |
| Past year experience of sexual IPV | 31 (21%) | 8 (16%) | 35 (19%) | 74 (20%) | 21 (16%) | 10 (27%) | 51 (21%) | 82 (20%) |

## What factors are associated with women's past year IPV?

Overall, 29% of women whose male partner completed the survey reported having experienced physical and/or sexual IPV by that partner in the past year (18% physical, 20% sexual).

**Childhood factors (Tables 5 and 6).** Women's experiences of abuse as a child (witnessing violence against a household member, and/or experiencing sexual abuse or severe physical abuse themselves) were strongly associated with past year odds of physical (aOR 1.80, 95%CI 0.99–3.28) and sexual (aOR 1.78, 95%CI 1.03–3.10) IPV. Men's experiences of abuse as a child were not associated with women's reports of either type of IPV.

In the crude analysis, women's secondary education (and to a lesser extent men's) was associated with increased odds of sexual IPV. These variables were omitted from the multivariate analysis in favour of a relationship-level relative education variable (see below).

**Current demographics.** In the crude analysis, both men's and women's age were associated with past year-IPV. For women, odds of both types of IPV decreased consistently with age, with odds of sexual IPV falling sharply in those aged 50 or over (Tables 6 and 7). Men's older age was also associated with decreased IPV among women, though sexual IPV declined at an older age (50+) than physical IPV. The association with the man's age remained in the multivariate analysis (woman's age was excluded because it was highly correlated with man's age).

Disparity in education was also associated with increased odds of IPV, particularly sexual IPV. Compared to couples where neither had above primary education, odds of sexual IPV were more than doubled in relationships where one partner was educated above primary level and the other was not (aOR 3.19, 95%CI 1.66–6.14 where just she has it; aOR 2.30, 95%CI 1.20–4.41 where just he has it).

Neither the man's income alone nor the couple's combined income was related to either type of IPV. However, women's increasing income was related to decreased odds of physical IPV (comparing highest quartile to lowest, aOR 0.34, 95%CI 0.16–0.74). Physical IPV was also lower where neither the man nor woman had reported household-level financial hardship, though the confidence interval was wide and included unity in the adjusted analysis (aOR 0.39, 95%CI 0.13–1.14).

There was no association between either type of IPV and duration of relationship, the man having other wives, or the presence of non-biological children (of the man) in the household.

**Table 4. Descriptive data on respondents to male survey, disaggregated by trial arm.**

| | Control | Intervention | Total |
|---|---|---|---|
| | N = 180 | N = 245 | N = 425 |
| | n (%) | n (%) | n (%) |
| **Socio-demographics** | | | |
| Age (yrs) *Mean (sd)* | *49.8 (10.6)* | *46.7 (10.9)* | *48.0 (10.9)* |
| *<40* | 26/180 (14%) | 72/245 (29%) | 98/425 (23%) |
| *40–49* | 72 (40%) | 84 (34%) | 156 (37%) |
| *50+* | 82 (46%) | 89 (36%) | 171 (40%) |
| Marital status | | | |
| *Not married/living together* | 3 (2%) | 2 (1%) | 5 (1%) |
| *Married* | 162 (90%) | 219 (89%) | 381 (90%) |
| *Living together as if married* | 15 (8%) | 24 (10%) | 39 (9%) |
| Relationship duration | | | |
| *<5yrs* | 10 (6%) | 16 (7%) | 26 (6%) |
| *5–9.99 yrs* | 14 (8%) | 35 (14%) | 49 (12%) |
| *10+ yrs* | 156 (87%) | 194 (79%) | 350 (82%) |
| Has other wife/wives | 14/179 (8%) | 32/244 (13%) | 46/423 (11%) |
| Has biological children | 178 (99%) | 242 (99%) | 420 (99%) |
| All biological children are from this partner | 91/178 (51%) | 120/242 (50%) | 211/420 (50%) |
| Non-biological children living in household | 91 (51%) | 116 (47%) | 207 (49%) |
| Above primary education | 63/180 (35%) | 65/245 (27%) | 128/425 (30%) |
| Employment | | | |
| *Hasn't earned* | 5/180 (3%) | 8/245 (3%) | 13/425 (3%) |
| *Self-employed* | 136 (76%) | 192 (78%) | 328 (77%) |
| *Works for someone else* | 36 (20%) | 41 (17%) | 77 (18%) |
| *Self-employed and works for someone else* | 3 (2%) | 4 (2%) | 7 (2%) |
| Income quartile | | | |
| *1st quartile (lowest)* | 43 (24%) | 66 (27%) | 109 (26%) |
| *2nd quartile* | 38 (21%) | 60 (24%) | 98 (23%) |
| *3rd quartile* | 55 (31%) | 68 (28%) | 123 (29%) |
| *4th quartile (highest)* | 39 (22%) | 43 (18%) | 82 (19%) |
| *Doesn't earn* | 5 (3%) | 8 (3%) | 13 (3%) |
| Household experienced financial hardship in past yr | 135 (75%) | 190 (78%) | 325 (76%) |
| **Health and risk behaviours** | | | |
| Poor mental health | 68 (38%) | 90 (37%) | 158 (37%) |
| Frequency of drinking alcohol in past week | | | |
| *Never* | 104 (58%) | 158 (64%) | 262 (62%) |
| *<2–3 times per week* | 43 (24%) | 54 (22%) | 97 (23%) |
| *2–3 times per week or more* | 33 (18%) | 33 (13%) | 66 (16%) |
| **Attitudes** | | | |
| Attitudes supporting of IPV | 59 (33%) | 82 (33%) | 141 (33%) |
| **Childhood experiences of violence** | | | |
| Beaten as a child at least a few times | 78 (43%) | 92 (38%) | 170 (40%) |
| Ever severely beaten as a child (injured/left marks) | 28 (16%) | 32 (13%) | 60 (14%) |
| Experienced sexual abuse as a child | 22 (12%) | 22 (9%) | 44 (10%) |

**Attitudes/behaviours/health.** Odds of both types of IPV were higher where men had attitudes supportive of IPV, though the associations were not statistically significant at the 5%

**Table 5. Age-adjusted odds ratios of association between childhood factors and women's past year experience of IPV (Blue = Man's characteristics; Yellow = Woman's characteristics).**

| | | Past year physical IPV | Age-adjusted OR (95%CI) | Past year sexual IPV | Age-adjusted OR (95%CI) |
|---|---|---|---|---|---|
| Man witnessed violence against a household member as a child | Never | 31/192 (16%) | - | 37/192 (19%) | - |
| | Ever | 46/228 (20%) | 1.24 (0.80–1.91) | 47/228 (21%) | 0.98 (0.64–1.51) |
| Man experienced sexual abuse or severe physical abuse as a child | Never | 58/332 (17%) | - | 63/332 (19%) | - |
| | Ever | 19/88 (22%) | 1.14 (0.65–2.00) | 21/88 (24%) | 1.22 (0.67–2.22) |
| Man witnessed violence against a household member as a child, or experienced sexual abuse or severe physical abuse herself as a child | Never | 28/168 (17%) | - | 33/168 (20%) | - |
| | Ever | 49/252 (19%) | 1.14 (0.72–1.80) | 51/252 (20%) | 0.92 (0.61–1.39) |
| Man's education | Primary or below | 57/295 (19%) | - | 54/295 (18%) | - |
| | Above primary | 20/125 (16%) | 0.87 (0.49–1.55) | 30/125 (24%) | 1.58 (0.95–2.62) |
| Woman witnessed violence against a household member as a child | Never | 23/146 (16%) | - | 21/146 (14%) | - |
| | Ever | 52/264 (20%) | 1.27 (0.77–2.10) | 59/264 (22%) | 1.67 (1.02–2.74) |
| Woman experienced sexual abuse or severe physical abuse as a child | Never | 46/287 (16%) | - | 49/287 (17%) | - |
| | Ever | 29/123 (24%) | 1.51 (0.97–2.37) | 31/123 (25%) | 1.50 (0.84–2.66) |
| Woman witnessed violence against a household member as a child, or experienced sexual abuse or severe physical abuse herself as a child | Never | 14/117 (12%) | - | 15/117 (13%) | - |
| | Ever | 61/293 (21%) | 1.80 (0.99–3.28) | 65/293 (22%) | 1.78 (1.03–3.10) |
| Woman's education | Primary or below | 60/327 (18%) | - | 57/327 (17%) | - |
| | Above primary | 15/83 (18%) | 0.93 (0.50–1.73) | 23/83 (28%) | 1.80 (1.00–3.22) |

level (aOR 1.42, 95%CI 0.78–2.58 for physical IPV; aOR 1.53, 95%CI 0.84–2.79 for sexual IPV) (Tables 6 and 8). Women's attitudes supportive of physical IPV were associated with increased odds of physical IPV (aOR 1.79, 95%CI 1.03–3.08) but not sexual IPV.

Frequent alcohol use among men was associated with both physical and sexual IPV, with women whose partners drank at least 2–3 times per week having an approximately two-fold increase in odds of both types of IPV, compared to those whose partners never drank alcohol. The 95% confidence interval excluded unity for sexual, though not physical IPV. While men's poor mental health was not associated with either physical or sexual IPV, women's poor mental health was associated with increased odds of sexual IPV (aOR 2.49, 95%CI 1.47–4.22) but not physical IPV.

## Discussion

This paper yields findings that are both methodological and substantive in nature. Firstly, we successfully conducted a cross-sectional survey of male partners of women enrolled in an IPV prevention trial, with the full consent of women participants. Secondly, we found important differences between women consenting for their partners to be approached in intervention and control arms, particularly with respect to IPV experience–past-year IPV experience was associated with increased consent rates in the intervention arm, but decreased consent in the control arm. Thirdly, we identified factors associated with IPV that pertain to the woman, her partner and the relationship more broadly. In this sample, factors associated with increased odds of physical IPV included the woman's childhood experiences of abuse, young age,

**Table 6. Results of multivariate analysis of risk factors for women's past year experience of IPV (Blue = Man's characteristics; Yellow = Woman's characteristics; Green = Relationship/household characteristics).**

| | | Past year physical IPV | Past year sexual IPV |
|---|---|---|---|
| **Childhood factors** | | Model 1 aOR* | Model 1 aOR* |
| | | (95%CI) | (95%CI) |
| Woman witnessed violence against a household member as a child, or experienced sexual abuse or severe physical abuse herself as a child | Never | - | - |
| | Ever | 1.80 (0.99–3.28) | 1.78 (1.03–3.10) |
| **Current demographics** | | Model 2 aOR** | Model 2 aOR** |
| | | (95%CI) | (95%CI) |
| Man's age | 19–39 yrs | - | - |
| | 40–49 yrs | 0.62 (0.32–1.22) | 1.39 (0.73–2.63) |
| | 50+ yrs | 0.31 (0.16–0.60) | 0.38 (0.18–0.82) |
| Woman's income quartile | 1st quartile (lowest) | - | - |
| | 2nd quartile | 0.91 (0.45–1.87) | 1.45 (0.77–2.75) |
| | 3rd quartile | 0.63 (0.30–1.31) | 1.79 (0.91–3.52) |
| | 4th quartile (Highest) | 0.34 (0.16–0.74) | 0.92 (0.43–1.96) |
| | Doesn't earn | 1.30 (0.54–3.11) | 1.63 (0.60–4.43) |
| | Don't know | 0.96 (0.33–2.85) | 0.86 (0.26–2.90) |
| Relative education status | Neither has above primary | - | - |
| | Just she does | 1.69 (0.75–3.82) | 3.19 (1.66–6.14) |
| | Just he does | 1.29 (0.66–2.53) | 2.30 (1.20–4.41) |
| | Both do | 0.54 (0.17–1.67) | 1.75 (0.78–3.94) |
| Either partner reported financial hardship (past yr) | Either/both did | - | - |
| | Neither did | 0.39 (0.13–1.14) | 0.86 (0.39–1.93) |
| **Attitudes and health** | | Model 3 aOR*** | Model 3 aOR*** |
| | | (95%CI) | (95%CI) |
| Man's attitudes on IPV | Never acceptable | - | - |
| | Ever acceptable | 1.42 (0.78–2.58) | 1.53 (0.84–2.79) |
| Man's alcohol use (any) | Doesn't drink | - | - |
| | <2–3 times per week | 1.37 (0.62–3.05) | 1.73 (0.95–3.17) |
| | 2–3 times per week or more | 2.00 (0.85–4.72) | 2.14 (1.14–4.02) |
| Woman's attitudes on IPV | Never acceptable | - | - |
| | Ever acceptable | 1.79 (1.03–3.08) | 0.77 (0.42–1.40) |
| Woman's poor mental health | No | - | - |
| | Yes | 1.19 (0.74–1.91) | 2.49 (1.47–4.22) |
| Woman's alcohol use (any) | Doesn't drink | - | - |
| | Drinks alcohol | 1.55 (0.89–2.70) | 0.61 (0.29–1.30) |

*Adjusted for age

**Adjusted for all 'Childhood factors' and 'Current demographics' variable in this table

***Adjusted for all 'Childhood factors', 'Current demographics' and 'Attitudes and health' variables in this table.

woman's lower income, and woman's attitudes towards IPV. Factors associated with increased odds of sexual IPV were the woman's childhood experiences of abuse, young age, disparity in education, partner's alcohol use and woman's poor mental health.

Most research into IPV has involved either women only or men only. There is a paucity of research that has included both members of a couple. We have shown that it is possible to

**Table 7. Age-adjusted odds ratios of association between current demographics and women's past year experience of IPV (Blue = Man's characteristics; Yellow = Woman's characteristics; Green = Relationship/household characteristics).**

| | | Past year physical IPV | | Past year sexual IPV | |
|---|---|---|---|---|---|
| | | | Age-adjusted OR (95%CI) | | Age-adjusted OR (95%CI) |
| Man's age | 19–39 yrs | 28/94 (30%) | - | 22/94 (23%) | - |
| | 40–49 yrs | 30/156 (19%) | 0.56 (0.30–1.05) | 44/156 (28%) | 1.29 (0.72–2.29) |
| | 50+ yrs | 19/170 (11%) | 0.30 (0.16–0.55) | 18/170 (11%) | 0.39 (0.19–0.79) |
| Man's income quartile | 1st quartile (lowest) | 17/107 (16%) | - | | - |
| | 2nd quartile | 24/96 (25%) | 1.70 (0.81–3.57) | 20/107 (19%) | 1.32 (0.69–2.51) |
| | 3rd quartile | 20/123 (16%) | 0.97 (0.41–2.29) | 25/96 (26%) | 0.71 (0.37–1.38) |
| | 4th quartile (Highest) | 15/81 (19%) | 1.18 (0.50–2.82) | 20/123 (16%) | 1.26 (0.65–2.43) |
| | Doesn't earn | 1/13 (8%) | 0.71 (0.09–5.78) | 19/81 (23%) | --- |
| Woman's age | <40 yrs | 45/195 (23%) | - | 49/195 (25%) | - |
| | 40–49 yrs | 21/143 (15%) | 0.57 (0.34–0.96) | 30/143 (21%) | 0.79 (0.49–1.27) |
| | 50+ yrs | 9/72 (13%) | 0.48 (0.22–1.04) | 1/72 (1%) | 0.04 (0.01–0.24) |
| Woman's income quartile | 1st quartile (lowest) | 25/109 (23%) | - | 17/109 (16%) | - |
| | 2nd quartile | 18/80 (23%) | 1.01 (0.50–2.03) | 18/80 (23%) | 1.52 (0.77–2.99) |
| | 3rd quartile | 15/95 (16%) | 0.65 (0.33–1.28) | 25/95 (26%) | 1.97 (1.05–3.69) |
| | 4th quartile (highest) | 8/92 (9%) | 0.33 (0.16–0.69) | 15/92 (16%) | 1.00 (0.49–2.04) |
| | Doesn't earn | 7/25 (28%) | 1.35 (0.61–2.96) | 6/25 (24%) | 1.70 (0.65–4.47) |
| | Don't know | 4/19 (21%) | 0.91 (0.32–2.58) | 3/19 (16%) | 1.02 (0.32–3.29) |
| Duration of relationship | 10+ yrs | 60/349 (17%) | - | 70/349 (20%) | - |
| | 5–9.9 yrs | 13/49 (27%) | 0.99 (0.47–2.11) | 9/49 (18%) | 0.61 (0.28–1.32) |
| | <5 yrs | 4/22 (18%) | 0.64 (0.24–1.70) | 5/22 (23%) | 0.83 (0.26–2.62) |
| Man has other wife/wives | No | 69/375 (18%) | - | 76/375 (20%) | - |
| | Yes | 8/45 (18%) | 0.98 (0.43–2.23) | 8/45 (18%) | 0.89 (0.36–2.19) |
| Age-gap | Man >5yrs older | 37/252 (15%) | - | 41/252 (16%) | - |
| | Same age/ older | 38/158 (24%) | 1.29 (0.73–2.28) | 39/158 (25%) | 1.41 (0.87–2.29) |
| Relative education status | Neither has above primary | 44/242 (18%) | - | 36/242 (15%) | - |
| | Just she does | 11/44 (25%) | 1.44 (0.65–3.18) | 14/44 (32%) | 2.93 (1.47–5.82) |
| | Just he does | 16/85 (19%) | 1.20 (0.63–2.27) | 21/85 (25%) | 2.29 (1.22–4.30) |
| | Both do | 4/39 (10%) | 0.52 (0.17–1.55) | 9/39 (23%) | 1.68 (0.71–3.95) |
| Children living in household that have a different biological father | No | 45/216 (21%) | - | 45/216 (21%) | - |
| | Yes | 32/204 (16%) | 0.81 (0.47–1.39) | 39/204 (19%) | 1.01 (0.56–1.85) |
| Combined monthly income quartile | 1st quartile (lowest) | 21/101 (21%) | - | 17/101 (17%) | - |
| | 2nd quartile | 25/106 (24%) | 1.09 (0.53–2.25) | 27/106 (25%) | 1.40 (0.74–2.66) |
| | 3rd quartile | 14/103 (14%) | 0.58 (0.26–1.30) | 22/103 (21%) | 1.21 (0.62–2.36) |
| | 4th quartile (highest) | 13/91 (14%) | 0.62 (0.29–1.34) | 15/91 (16%) | 0.85 (0.47–1.54) |
| | Don't know | 4/19 (21%) | 0.97 (0.31–3.05) | 3/19 (16%) | 0.85 (0.26–2.80) |
| Either partner reported household-level financial hardship in past year | Either/both did | 73/368 (20%) | - | 75/368 (20%) | - |
| | Neither did | 4/52 (8%) | 0.33 (0.12–0.95) | 9/52 (17%) | 0.85 (0.37–1.96) |

conduct research with both members of a couple, in a way that prioritises the woman's safety and gives her full control over whether or not her partner is invited to participate. However, the resulting sample may be biased in several ways. Women who consented for their partners to be

**Table 8. Age-adjusted odds ratios of association between attitudes and health variables and women's past year experience of IPV (Blue = Man's characteristics; Yellow = Woman's characteristics; Green = Relationship/household characteristics).**

| | | Past year physical IPV | Age-adjusted OR (95%CI) | Past year sexual IPV | Age-adjusted OR (95%CI) |
|---|---|---|---|---|---|
| Man's attitudes on IPV | Never acceptable | 44/281 (16%) | - | 50/281 (18%) | - |
| | Ever acceptable | 33/139 (24%) | 1.64 (0.98–2.72) | 34/139 (24%) | 1.50 (0.91–2.49) |
| Man's poor mental health | No | 46/264 (17%) | - | 50/264 (19%) | - |
| | Yes | 31/156 (20%) | 1.06 (0.68–1.66) | 34/156 (22%) | 1.12 (0.69–1.83) |
| Man's alcohol use (any) | Doesn't drink | 38/259 (15%) | - | 40/259 (15%) | - |
| | <2–3 times per week | 23/96 (24%) | 1.69 (0.84–3.40) | 23/96 (24%) | 1.71 (0.98–2.96) |
| | 2–3 times per week or more | 16/65 (25%) | 1.94 (0.90–4.18) | 21/65 (32%) | 2.47 (1.32–4.62) |
| Woman's attitudes on IPV | Never acceptable | 31/217 (14%) | - | 48/217 (22%) | - |
| | Ever acceptable | 46/203 (23%) | 1.77 (1.08–2.93) | 36/203 (18%) | 0.74 (0.43–1.28) |
| Woman's poor mental health | No | 40/250 (16%) | - | 36/250 (14%) | - |
| | Yes | 37/170 (22%) | 1.56 (0.98–2.46) | 48/170 (28%) | 2.40 (1.50–3.85) |
| Woman's alcohol use (any) | Doesn't drink | 54/326 (17%) | - | 65/326 (20%) | - |
| | Drinks alcohol | 23/94 (24%) | 1.54 (0.94–2.51) | 19/94 (20%) | 0.98 (0.54–1.76) |

approached differed to those who refused–hence, men participating in the survey were disproportionately more likely to be in more long-term and established relationships characterised by better relationship dynamics, compared to the wider pool of male partners of trial participants. It is possible that women in newer relationships feel less comfortable being associated with demands on their partner's time. Women in relationships with poorer communication might also be less likely to have told their partner about their own participation in MAISHA or be concerned that he would react negatively to her sharing his details with the study team. These findings highlight the risk that dyad research, and indeed IPV interventions that seek to work with couples/dyads, may tend to underrepresent those most likely to be affected by IPV [28, 29].

To a small degree, this underrepresentation was indeed the pattern we observed in the control arm of this study. As we would have expected, women with recent experience of physical/sexual IPV were less likely than woman without recent IPV experience to consent for the team to approach their partners. We hypothesise that this was due to women's concerns about their own safety if their violent partners found out about their involvement in the trial or became aggravated by questions about their relationships. However, the reverse pattern was seen in the intervention arm, where women with recent experience of physical/sexual IPV were more likely to consent than women with no recent IPV experience. It is possible that in intervention groups, women in violent relationships viewed their partner's interaction with the research team as a positive prospect, and one that might help their relationship. In other words, this could be indicative of a positive intervention 'effect'.

Overall, women's consent for their partner to be approached was higher in the intervention arm compared to control arm. Possible reasons for this are that women in the intervention arm felt more predisposed towards research involvement because of their own experiences of the intervention, that they felt safer and more positive about their relationships/partners as a result of the intervention, and that they believed their relationships may benefit from their partner's engagement with the research team. Indeed, anecdotal evidence suggests that women in intervention groups were eager for their partners to receive a similar kind of intervention.

As well as highlighting the challenges involved in engaging couples affected by IPV in IPV-related research and prevention interventions, the findings illustrate the potential pitfalls of

using couples' data from men to estimate intervention impacts on male outcomes. The fact that perpetrators were overrepresented in the intervention arm and underrepresented in the control arm precludes using these data to estimate intervention impacts on male outcomes, such as gender attitudes, that might be related to IPV. Even if both members of a couple were recruited into a study from the outset, prior to randomisation, attrition could introduce differential selection bias between the study arms over the course of a trial. These selection biases must also be borne in mind when interpreting the results of our risk factor analyses.

As has been found in many settings [7, 30–32], women's childhood experiences of abuse (witnessing violence against a household member, or experiencing physical and/or sexual abuse themselves) were strongly associated with both physical and sexual IPV. Social Learning Theory [33] has been widely applied to the study of the 'intergenerational transmission' of violence, and posits that children who witness IPV against their mothers learn to normalise IPV and are more likely to imitate or tolerate such behaviours in their own relationships. Some evidence suggests that children's own experiences of abuse (that often co-occur with witnessing IPV) may be even more important drivers of later IPV risk [34]. Jewkes et al. [35] performed structural equation modelling using data from four countries to explore pathways between women's experiences of childhood trauma and later experiences of IPV. As well as having a direct influence on later IPV, they found childhood trauma influenced women's subsequent selection of a partner, decreasing the likelihood that she had a low-alcohol using partner. Women who had been exposed to childhood trauma were also more likely to have conservative attitudes towards gender equity.

Contrary to findings from other studies [13, 14, 30, 34], men's childhood experiences of abuse were not associated with women's past-year experience of either physical or sexual IPV in this sample. While we focused on childhood experience of severe physical abuse (that left marks or injuries), we observed similar null findings in relation to a more general measure of physical abuse in childhood (including less severe incidents). It is unclear why we did not observe an association in this study. Pathways through which childhood trauma may influence later IPV perpetration include normalisation of IPV and increased likelihood of alcohol misuse [30], though in our data we observed no association between childhood experiences of abuse and either current alcohol use or attitudes accepting of IPV. It is possible that reporting bias played a role in this null finding. In addition to stigma (and conversely normalisation), which may lead to underreporting of childhood abuse, it is also worth noting that we asked about witnessing violence against a household member, rather than witnessing parental IPV specifically. It is also possible that the selection bias present in this sample, towards more stable relationships with better relationship dynamics (potential mediators of the association between childhood abuse and IPV perpetration), has masked an association that would have been observed in the overall pool of male partners of trial participants.

It is also worth considering whether findings from other studies on the association between childhood abuse and later IPV perpetration could be somewhat an artefact of how the data are collected. Where women are asked to report on their male partner's childhood experiences of abuse, it is possible that they are more likely to know of the abuse in cases where it is most severe, or that they disproportionately know about their partner's experiences of abuse when they themselves are in a violent relationship. On the other hand, in studies conducted with men, it is possible that violent men (who admit to using IPV) may be more likely to recall/report childhood abuse than men who do not use/ admit to using IPV [34]. In our study, data on men's childhood experiences were collected from the men in the absence of questions on IPV perpetration. IPV data were collected from women only. Therefore, such differential reporting bias will not have occurred.

Our findings on the association between young age and increased IPV are consistent with other studies [7, 15]. It is interesting to note that odds of physical IPV declines steadily with the man's age, while risk of sexual IPV only decreases in the oldest age category for men (50 + years). One hypothesis to explain this is that sexual violence might be less dependent on situational factors, such as binge-drinking episodes and quarrelling, that tend to decline steadily with age, and more a persistent normative feature of relationships. It also has implications in terms of age-targeting of violence prevention programmes, which are typically targeted towards younger men.

We did not find any association between the man's income quartile or the couple's combined income and either type of IPV. While many studies have found an association between poverty/lower socioeconomic status and increased IPV risk [7, 8, 36], the association is not found in all settings [14]. It is possible that the income distribution in this study is too limited for an association to be observed–microfinance is specifically targeted towards poor women. We did, however, observe an association between woman's income quartile and physical (though not sexual) IPV. As well as reducing extreme levels of financial hardship, women's income can reduce women's economic dependence on their partners and increase their power/confidence within the relationship. In the opposite causal direction, lack of IPV can also lead to increased earning potential. This finding, alongside the converse potential for women's greater financial autonomy to increase risk of relationship conflict if it is perceived as undermining the male partner's authority or transgressing women's traditional gender norms, is discussed in more detail elsewhere [37].

Higher education was also associated with a slight increase in risk of sexual IPV, in contrast to baseline where more highly educated women had lower risk of IPV [38]. The greatest increase in risk was observed where there was a disparity in educational attainment between the woman and man. This pattern of risk has been observed in several other settings [7], with educational disparity potentially indicative of broader unequal power relations within the relationship (where the man is more highly educated), or perceived as a transgression of gender norms (where the woman is more highly educated).

As has been found in other studies [6, 7, 20], women's attitudes accepting of IPV were related to her risk of physical IPV. Surprisingly, in our multivariate analysis, women's attitudes were more strongly related to IPV risk than men's attitudes were. Men's permissive attitudes towards IPV have been found to be strongly associated with IPV perpetration by other studies, though not in all settings [14, 20]. It is possible that the lack of a strong association in our sample is due to social desirability bias in men's reporting of attitudes, which might be exacerbated by their knowledge of their partner's involvement in an intervention on healthy relationships. Upon further investigation of our data, we also found that the inclusion of alcohol in the multivariate model somewhat attenuated the association between men's attitudes and physical IPV, a pattern that Hindin and Kishor similarly observed in their analysis of IPV data from Rwanda [20]. Alcohol use by men in this sample was a strong risk factor for IPV, in keeping with a large body of evidence [7, 8, 39]. The complex relationships and interactions between alcohol, attitudes and IPV are now coming under scrutiny in an attempt to better understand how and when alcohol use leads to IPV perpetration [40, 41].

While women's poor mental health was strongly associated with IPV, we observed no association between men's poor mental health and IPV, in contrast to other studies [8, 14]. Again, this could be a result of selection bias, with those relationships suffering from the greatest strain (poor mental health and IPV) potentially excluded from our sample.

It is interesting to note that, although we found physical and sexual IPV to share some common risk factors, they do not share all. There is a slightly different age profile for each, while certain factors–poverty, women's lower income, and women's attitudes permissive of IPV–are

associated with increased risk of physical but not sexual IPV. While the two types of violence often co-occur, this is not always the case either in this setting [38] or elsewhere [8]. Certain distal risk factors, such as structural gender inequality, may drive underlying risk of all types of IPV, but interventions should not be grounded on the assumption that all types of IPV are amenable to the same intervention strategies. Interventions often have more success in reducing physical IPV than sexual IPV [26, 42, 43], and this may be because they target (or have more success in changing) the more proximate stressors (e.g. financial stresses) and triggers (e.g. quarrelling, communication) that are more relevant to physical IPV. Fulu et al. argue that factors associated with sexual IPV perpetration, relating to norms of masculinity that emphasise men's sexual dominance over women, and toughness and dominance over other men, appear "more similar to those associated with non-partner sexual violence than those associated with physical IPV" [8, 44].

This study has some limitations. As has already been noted, this couples' sample is not representative of the partnerships of all women who took part in the MAISHA CRT. The results of the risk factor analysis cannot therefore be generalised to this wider population. Several measures are also likely affected by reporting bias—for example childhood experiences of abuse may be underreported due to stigma. To reduce the risk of reporting bias for the IPV outcome, we used women's reports of experience rather than men's reports of perpetration, and used questions that are standardised and widely used in violence research [4], administered by interviewers who had received extensive training in conducting research relating to VAW.

The cross-sectional analysis precludes any inference on the direction of associations or causality. However, we can be confident that several of the factors explored, such as education and childhood experiences of abuse, at least preceded the occurrence of IPV even if the associations are not causal. The staged modelling process also takes account of the hierarchical nature of the relationships between the risk factors and IPV.

Another limitation is that the risk factor analysis was restricted to physical and sexual IPV. Emotional and economic abuse are now also recognised as prevalent types of partner abuse with serious consequences for women's health and well-being [45]. Future analyses are planned to explore male factors associated with these types of abuse. It is a strength of the analysis, however, that we explored physical and sexual IPV separately so as to explore how risk profiles differ between the two types of violence.

Due to lack of data on community-level risk factors, we were unable to explore these. However, a strength of our study is having couples' data rather than data just pertaining to women or men. Furthermore, we have a wealth of data (from women) even in cases where the male partner wasn't interviewed, allowing us to assess the extent and implications of selection bias in our final couples' sample.

## Conclusions

This is one of few studies on IPV that has collected data from couples in Tanzania [13, 46, 47]. We have shown that it is feasible to collect data from couples, prioritising women's safety, within the context of an intervention trial in which the intervention is delivered solely to women. We found that participation in the MAISHA intervention was associated with increased willingness among women for their partner to be interviewed, an outcome that might be considered indicative of improvements in relationship dynamics in the intervention arm. Importantly, we have contributed to the growing body of literature regarding risk factors for IPV. The breadth of risk factors relating to both members of the couple, that remain significant even when included together in multivariate models, demonstrates the need for more IPV prevention interventions that work with both women and men, and specifically with

couples. Recognition that risk and protective factors differ for physical and sexual IPV is essential to ensure that intervention content and strategies address both forms of IPV.

## Supporting information

**S1 Questionnaire. Female participant follow-up questionnaire (English).**
(PDF)

**S2 Questionnaire. Female participant follow-up questionnaire (Swahili).**
(PDF)

**S3 Questionnaire. Male partner questionnaire (English).**
(DOCX)

**S4 Questionnaire. Male partner questionnaire (Swahili).**
(PDF)

**S1 Table. Variables used in analysis of factors associated with IPV.**
(DOCX)

**S2 Table.** (a) Odds ratios of association between women's/relationship characteristics and women's consent to invite partner for interview; (b) Odds ratios of association between women's experience of IPV and women's consent to invite partner for interview, disaggregated by trial arm.
(DOCX)

## Acknowledgments

We thank all study participants for their time and commitment to the trial. We also thank the MAISHA research team for their contribution and dedication to implementing the trial in Tanzania, and to the administration teams at the Mwanza Intervention Trials Unit and London School of Hygiene & Tropical Medicine for their support.

## Author Contributions

**Conceptualization:** Gerry Mshana, Shelley Lees, Saidi Kapiga, Sheila Harvey.

**Data curation:** Tanya Abramsky, Christian Holm Hansen, Ramadhan Hashim, Sheila Harvey.

**Formal analysis:** Tanya Abramsky.

**Funding acquisition:** Shelley Lees, Saidi Kapiga.

**Investigation:** Gerry Mshana, Shelley Lees, Saidi Kapiga, Sheila Harvey.

**Methodology:** Gerry Mshana, Shelley Lees, Saidi Kapiga, Sheila Harvey.

**Project administration:** Imma Kapinga, Sheila Harvey.

**Supervision:** Imma Kapinga, Saidi Kapiga, Sheila Harvey.

**Writing – original draft:** Tanya Abramsky, Sheila Harvey.

**Writing – review & editing:** Tanya Abramsky, Imma Kapinga, Gerry Mshana, Shelley Lees, Christian Holm Hansen, Ramadhan Hashim, Heidi Stöckl, Saidi Kapiga, Sheila Harvey.

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
