## [Decision Letter · Decision Letter 0]

9 Apr 2020

PONE-D-20-05439

Factors associated with intimate partner violence in north-western Tanzania: Results from a survey of male partners of women enrolled in the MAISHA cluster randomised trial

PLOS ONE

Dear Miss Abramsky,

Thank you for submitting your manuscript to PLOS ONE. After careful consideration, we feel that it has merit but does not yet fully meet PLOS ONE’s publication criteria as it currently stands. Therefore, we invite you to submit a revised version of the manuscript that addresses the points raised during the review process.

Please address or carefully rebut reviewer comments as per your best judgement, and ensure that you will be fully aligned with PLOS' data availability policy should the revised manuscript be accepted for publication. 

We would appreciate receiving your revised manuscript by May 24 2020 11:59PM. To enhance the reproducibility of your results, we recommend that if applicable you deposit your laboratory protocols in protocols.io, where a protocol can be assigned its own identifier (DOI) such that it can be cited independently in the future. For instructions see: http://journals.plos.org/plosone/s/submission-guidelines#loc-laboratory-protocols

We look forward to receiving your revised manuscript.

Kind regards,

Kristin Dunkle

Academic Editor

PLOS ONE

Journal Requirements:

3. Thank you for stating in your financial disclosure:  

"The MAISHA study was funded by an anonymous donor and supported by the STRIVE Research Consortium, which is funded by UK Aid (Grant number PO5244 held by SL) from the UK government Department for International Development (https://www.ukaiddirect.org/) . The views expressed in the paper do not necessarily reflect the Department’s official policies. The funders had no role in study design, data collection and analysis, decision to publish, or preparation of the manuscript. "

PLOS ONE requires you to include in your manuscript further information about the funder so that any relevant competing interests can be assessed. Please respond to the following questions:

a)    Please state whether any of the research costs or authors' salaries were funded, in whole or in part, by a tobacco company (our policy on tobacco funding is at http://journals.plos.org/plosone/s/disclosure-of-funding-sources)  

b)    Please state whether the donor has any competing interests in relation to this work (see http://journals.plos.org/plosone/s/competing-interests) .

c)    Please state whether the identity of the donor might be considered relevant to editors or reviewers’ assessment of the validity of the work.

d)    If the donors have no perceived or actual competing interests, please state: “The authors are not aware of any competing interests”.

This information should be included in your cover letter. We will amend your financial disclosure and competing interests on your behalf.

Additional Editor Comments (if provided):

This is a fine manuscript and we shall be pleased to reconsider it when you have addresed the reviewer's comments. Please note also PLOS' data availability policies and ensure that, absent any compelling competing concerns, the data will be available should the manuscript be published.

Reviewers' comments:

Reviewer's Responses to Questions

**Comments to the Author**

1. Is the manuscript technically sound, and do the data support the conclusions?

Reviewer #1: Yes

Reviewer #2: Partly

2. Has the statistical analysis been performed appropriately and rigorously? 

Reviewer #1: Yes

Reviewer #2: Yes

3. Have the authors made all data underlying the findings in their manuscript fully available?

Reviewer #1: No

Reviewer #2: No

4. Is the manuscript presented in an intelligible fashion and written in standard English?

Reviewer #1: Yes

Reviewer #2: Yes

5. Review Comments to the Author

Reviewer #1: Congratulations on a well written paper. My comments reflect suggestions for minor revisions.

Abstract: The abstract does not accurately reflect your most important findings. The first paragraph of the discussion nicely describes your three most important findings but you only present results from the third among these. Suggest revising the abstract to present the data most reflective of these three points. In the abstract you discuss man- woman- relationship- and household level factors. While Figure 1 aptly explains this combining relationship and household factors, in the full paper the relationship and household factors are never clearly delineated nor are the defined elsewhere in the paper. Are they the same or not. If not need to more clearly define in the full paper.

General: Data are plural. Do a control f and check grammar, "Data are/were" as opposed to "Data is" (ie page 10, line 66)

Background: Add delineation of social ecology (man, woman, relationship and household) here or otherwise define more clearly in the methods measures

Methods: methods are well done and clear. Add info on social ecology as relates to your specific measures in this section with the sub section on IPV outcomes

Results:

Add percents to the phrase "938 (%) were interviewed of whom 790 (%) reported..."

Table 1: extend spacing of column 1 so it reads more clearly. There is space in columns 2-4 to do this

Table 2a: define income quartiles and BRAC for readers unfamiliar with these terms

Table 3: add the word quartiles under income quartile so consistently labeled across tables 2a and 3

Do not begin sentences with a number (ie 30%... page 21, row 260)

Discussion: This is the most strongly written part of this paper and this strength should be reflected in the abstract as mentioned previously

Page 32, line 399-411 speak to measures of severity of violence experienced by men and whether this might have played a role in your finding of no association

Page 33, line 429-439 need to address how increased financial independence and/or women's increased autonomy may also trigger violence

Reviewer #2: Thank you for the opportunity to review this paper that is a valuable contribution to literature that focuses on the dynamics of conducting violence related research among couples. I have several comments to make which may be helpful to the authors

Title: Although the paper does look at factors associated with women’s victimisation and corroborates data from male partners, the current title misses the main objective and novelty of the paper which is the process and selection associated with women’s willingness to have male partners participating in research with them. It may be good to reflect this somehow in the title if possible.

Introduction: This is well written and has logical flow including a strong rationale for the study.

Methods:

It is clear that male partners were not asked about violence perpetration to minimise risks to female participants. However, in line 138 there’s reference that men were asked about relationship characteristics and dynamics. Male participants were also questioned about attitudes towards IPV. It will be important for authors to also discuss the potential for risk associated with administering such questions with male partners in context of them being abusive. It may be without asking about violence perpetration there is chance of suspicion by male partners who may well be guarded about their partners discussing their relationships with outsiders.

Results:

The authors report on insignificant associations i.e 95%CI overlapping with 1 as “weak associations”. This is problematic and must be rectified including discussions that allude to these as follows:

Line 320 Physical IPV is associated with financial hardship

Line 324 men’s attitudes are said to be weakly associated with physical and sexual IPV

Line 327 -typo- 95% CI from table is 1.03-3.08

Line 328- Men’s alcohol is associated with physical IPV

Line 331- There is “suggestion “that women’s alcohol is weakly associated with physical IPV

Overall authors must use the standard that every OR with 95%CI overlapping with 1 should not be considered as a significant weak association.

Discussion:

The discussion must be revised in accordance to changes in interpretation of results as alluded above.

The finding that men’s violence perpetration was not associated with their attitudes towards IPV warrants further discussion. Could that have been a result of social desirability biases or how else can this be explained.

The discussion does not provide explanation about what could be happening with women who did not disclose IPV but still were unwilling for their male partners to participate. These may be an important group to understand

Limitations

A missed opportunity is that researchers did not explore women’s motivations for their un/willingness to let their male partners participate in the study. It is important to cite this as a limitation. However, the speculated explanations based on data distribution of women who consented is well argued.

6. PLOS authors have the option to publish the peer review history of their article (what does this mean?). If published, this will include your full peer review and any attached files.

Reviewer #1: No

Reviewer #2: No

---

## [Author Response · Author response to Decision Letter 0]

21 May 2020

Reviewer #1: Congratulations on a well written paper. My comments reflect suggestions for minor revisions.

TA: THANK YOU FOR THIS POSITIVE FEEDBACK. 

Abstract: The abstract does not accurately reflect your most important findings. The first paragraph of the discussion nicely describes your three most important findings but you only present results from the third among these. Suggest revising the abstract to present the data most reflective of these three points. 

TA: THANK YOU FOR THIS COMMENT. IT IS ALWAYS A CHALLENGE TO SUMMARISE ALL OF THE KEY POINTS WITHIN THE STRICT WORD LIMIT OF AN ABSTRACT. WE CITE THE CONSENT RATES IN THE ABSTRACT’S RESULTS SECTION AND HAVE ADDED A SENTENCE TO THE ABSTRACT’S CONCLUSION STATING THAT WE SUCCESSFULLY CONDUCTED A SURVEY OF MALE PARTNERS WITH THE FULL CONSENT OF WOMEN. WE HAVE ALSO ADDED A SENTENCE TO THE RESULTS SECTION OF THE ABSTRACT REGARDING THE DIFFERENT PATTERNS OF ASSOCIATION OBSERVED IN THE TWO TRIAL ARMS WITH RESPECT TO WOMEN’S IPV EXPERIENCE AND CONSENT. 

In the abstract you discuss man- woman- relationship- and household level factors. While Figure 1 aptly explains this combining relationship and household factors, in the full paper the relationship and household factors are never clearly delineated nor are the defined elsewhere in the paper. Are they the same or not. If not need to more clearly define in the full paper.

TA: WE HAVE NOW ADDED A SUBSECTION TO THE METHODS SECTION OF THE FULL PAPER ENTITLED “EXPOSURE VARIABLES FOR THE IPV ANALYSIS”. THIS OUTLINES THE DIFFERENT LEVELS OF THE SOCIAL ECOLOGY CONSIDERED, AND REFERS TO THE CONCEPTUAL FRAMEWORK (FIGURE 1) WHERE VARIABLES ARE LAID OUT IN MORE DETAIL. IN THIS PARAGRAPH, WE CLARIFY THAT RELATIONSHIP/HOUSEHOLD FACTORS ARE “FACTORS PERTAINING TO THE RELATIONSHIP ITSELF AS WELL AS TO THE BROADER HOUSEHOLD IN WHICH THE COUPLE LIVES”. HERE, AS WELL AS IN THE ABSTRACT AND THROUGHOUT THE PAPER, RELATIONSHIP- AND HOUSEHOLD-LEVEL FACTORS ARE REFERRED TO AS “RELATIONSHIP-/HOUSEHOLD-LEVEL FACTORS” TO MAKE IT CLEAR THAT THEY ARE BEING CONSIDERED TOGETHER. 

General: Data are plural. Do a control f and check grammar, "Data are/were" as opposed to "Data is" (ie page 10, line 66)

TA: THANKS FOR POINTING THIS OUT. I HAVE MADE THE NECESSARY CHANGES THROUGHOUT THE MANUSCRIPT. 

Background: Add delineation of social ecology (man, woman, relationship and household) here or otherwise define more clearly in the methods measures

TA: SEE ABOVE COMMENT ON SOCIAL ECOLOGY 

Methods: methods are well done and clear. Add info on social ecology as relates to your specific measures in this section with the sub section on IPV outcomes 

TA: SEE ABOVE COMMENT ON SOCIAL ECOLOGY

Results:

Add percents to the phrase "938 (%) were interviewed of whom 790 (%) reported..."

TA: WE HAVE ADDED THESE PERCENTAGES AS REQUESTED. 

Table 1: extend spacing of column 1 so it reads more clearly. There is space in columns 2-4 to do this

TA: WE HAVE MADE THIS CHANGE TO TABLE 1 

Table 2a: define income quartiles and BRAC for readers unfamiliar with these terms

TA: WE HAVE CHANGED THE LABELS FOR THESE VARIABLES, AND ADDED FOOTNOTES TO TABLE 2 TO EXPLAIN BRAC AND INCOME QUARTILES. 

Table 3: add the word quartiles under income quartile so consistently labeled across tables 2a and 3

TA: WE HAVE MADE THIS CHANGE TO TABLES 3, 4B AND 5. 

Do not begin sentences with a number (ie 30%... page 21, row 260)

TA: WE HAVE FIXED THE THREE INSTANCES WHERE THIS OCCURRED ON PAGE 16. 

Discussion: This is the most strongly written part of this paper and this strength should be reflected in the abstract as mentioned previously 

TA: THE ABSTRACT WAS AMENDED TO MORE ACCURATELY SUMMARISE KEY POINTS FROM THE DISCUSSION, AS DESCRIBED IN RESPONSE TO YOUR FIRST COMMENT. 

Page 32, line 399-411 speak to measures of severity of violence experienced by men and whether this might have played a role in your finding of no association

TA: WE HAVE ADDED THE FOLLOWING SENTENCE TO THIS PARAGRAPH: “WHILE WE FOCUSED ON CHILDHOOD EXPERIENCE OF SEVERE PHYSICAL ABUSE (THAT LEFT MARKS OR INJURIES), WE OBSERVED SIMILAR NULL FINDINGS IN RELATION TO A MORE GENERAL MEASURE OF PHYSICAL ABUSE IN CHILDHOOD (INCLUDING LESS SEVERE INCIDENTS).”

Page 33, line 429-439 need to address how increased financial independence and/or women's increased autonomy may also trigger violence

TA: WE HAVE AMENDED THE SENTENCE AT THE END OF THIS PARAGRAPH TO READ: “THIS FINDING, ALONGSIDE THE CONVERSE POTENTIAL FOR WOMEN’S GREATER FINANCIAL AUTONOMY TO INCREASE RISK OF RELATIONSHIP CONFLICT IF IT IS PERCEIVED AS UNDERMINING THE MALE PARTNER’S AUTHORITY OR TRANSGRESSING WOMEN’S TRADITIONAL GENDER NORMS, IS DISCUSSED IN MORE DETAIL ELSEWHERE.”

**

Reviewer 2

Thank you for the opportunity to review this paper that is a valuable contribution to literature that focuses on the dynamics of conducting violence related research among couples. I have several comments to make which may be helpful to the authors

Title: Although the paper does look at factors associated with women’s victimisation and corroborates data from male partners, the current title misses the main objective and novelty of the paper which is the process and selection associated with women’s willingness to have male partners participating in research with them. It may be good to reflect this somehow in the title if possible.

TA: WE HAVE CHANGED THE TITLE TO: 

COUPLES DATA FROM NORTH-WESTERN TANZANIA: INSIGHTS FROM A SURVEY OF MALE PARTNERS OF WOMEN ENROLLED IN THE MAISHA CLUSTER RANDOMIZED TRIAL OF AN INTIMATE PARTNER VIOLENCE PREVENTION INTERVENTION

Introduction: This is well written and has logical flow including a strong rationale for the study.

TA: Thank you for this positive feedback. 

Methods:

It is clear that male partners were not asked about violence perpetration to minimise risks to female participants. However, in line 138 there’s reference that men were asked about relationship characteristics and dynamics. Male participants were also questioned about attitudes towards IPV. It will be important for authors to also discuss the potential for risk associated with administering such questions with male partners in context of them being abusive. It may be without asking about violence perpetration there is chance of suspicion by male partners who may well be guarded about their partners discussing their relationships with outsiders. 

TA: THIS IS AN IMPORTANT POINT. WE HAVE ADDED A COUPLE OF SENTENCES TO THE BOTTOM OF THIS PARAGRAPH ON HOW QUESTIONS ON ATTITUDES AND RELATIONSHIP DYNAMICS WERE FRAMED TO MINIMISE RISK TO FEMALE PARTICIPANTS: “QUESTIONS ON ATTITUDES TOWARDS IPV WERE EMBEDDED IN A SECTION ON BROADER NORMS AND ATTITUDES (INCLUDING EQUITABLE AS WELL AS CONSERVATIVE NORMS), AND INTRODUCED IN NEUTRAL TERMS, SO AS NOT TO CAST MEN AS THE AGGRESSORS WITHIN RELATIONSHIPS. IN THE SAME VEIN, QUESTIONS ON RELATIONSHIP DYNAMICS COVERED POSITIVE AS WELL AS NEGATIVE INTERACTIONS WITHIN THE COUPLE.” 

IT IS ALSO IMPORTANT TO EMPHASISE THAT, AS DESCRIBED IN THE METHODS SECTION, WOMEN WERE MADE AWARE OF THE NATURE OF QUESTIONS IN THE MALE SURVEY BEFORE GIVING CONSENT FOR THE TEAM TO APPROACH THEIR PARTNER. CONCERNS ABOUT RISK POSED BY SUCH QUESTIONS IN THE CONTEXT OF SPECIFIC RELATIONSHIPS WERE NO DOUBT BEHIND SOME WOMEN’S DECISIONS TO WITHHOLD CONSENT, AS HAS BEEN DISCUSSED IN THE PAPER. 

Results:

The authors report on insignificant associations i.e 95%CI overlapping with 1 as “weak associations”. This is problematic and must be rectified including discussions that allude to these as follows:

Line 320 Physical IPV is associated with financial hardship

Line 324 men’s attitudes are said to be weakly associated with physical and sexual IPV

Line 328- Men’s alcohol is associated with physical IPV

Line 331- There is “suggestion “that women’s alcohol is weakly associated with physical IPV

Overall authors must use the standard that every OR with 95%CI overlapping with 1 should not be considered as a significant weak association. 

TA: THERE IS INCREASING RECOGNITION THAT THE IMPORTANCE OR RELEVANCE OF AN OBSERVED ASSOCIATION SHOULD NOT BE ACCEPTED OR DISMISSED SOLELY ON THE BASIS OF WHETHER IT IS STATISTICALLY SIGNIFICANT AT THE 5% LEVEL. NOT ONLY IS THIS A SOMEWHAT ARBITRARY CUT-OFF POINT (WITH GRADED INFERENCE FROM THE ACTUAL P-VALUE POTENTIALLY MORE USEFUL), BUT IT IS ALSO LARGELY INFLUENCED BY SAMPLE SIZE AND THE CONSEQUENT POWER OF THE STUDY TO DETECT ASSOCIATIONS OF ANY GIVEN SIZE. WHILE P-VALUES AND CONFIDENCE INTERVALS ARE OBVIOUSLY CRUCIAL TO EVALUATE THE STRENGTH OF EVIDENCE, IT IS ALSO IMPORTANT TO CONSIDER THE DIRECTION AND MAGNITUDE OF AN OBSERVED ASSOCIATION. IN RELATION TO THE FINDINGS YOU MENTION ABOVE, WE HAVE CHANGED THE WORDING TO MAKE IT CLEAR WHERE CONFIDENCE INTERVALS INCLUDE UNITY – HOWEVER, WE STILL DRAW ATTENTION TO OBSERVED PATTERNS IN THE DATA WHERE RELATIVELY LARGE ODDS RATIOS SUGGEST THAT ODDS OF IPV ARE INCREASED IN ONE EXPOSURE CATEGORY COMPARED TO ANOTHER. FOR EXAMPLE: “PHYSICAL IPV WAS ALSO LOWER WHERE NEITHER THE MAN NOR WOMAN HAD REPORTED HOUSEHOLD-LEVEL FINANCIAL HARDSHIP, THOUGH THE CONFIDENCE INTERVAL WAS WIDE AND INCLUDED UNITY IN THE ADJUSTED ANALYSIS (AOR 0.39, 95%CI 0.13-1.14).”

WE HAVE REMOVED MENTION OF MEN’S ALCOHOL AS A RISK FACTOR FOR PHYSICAL IPV FROM THE ABSTRACT WHERE MORE DETAILED DISCUSSION OF DIRECTION VERSUS STATISTICAL SIGNIFICANCE IS NOT POSSIBLE. 

Line 327 -typo- 95% CI from table is 1.03-3.08

TA: THANKS FOR POINTING OUT THE TYPO. THIS HAS NOW BEEN CORRECTED.

Discussion: 

The discussion must be revised in accordance to changes in interpretation of results as alluded above.

TA: SEE RESPONSE TO ABOVE COMMENT.

The finding that men’s violence perpetration was not associated with their attitudes towards IPV warrants further discussion. Could that have been a result of social desirability biases or how else can this be explained. 

TA: THIS IS AN IMPORTANT COMMENT. WE HAD ALREADY INCLUDED SOME DISCUSSION OF HOW THE INCLUSION OF ALCOHOL IN THE MODEL FURTHER ATTENUATED OBSERVED ASSOCIATIONS BETWEEN ATTITUDES AND IPV. WE HAVE ALSO ADDED THE FOLLOWING SENTENCE TO THE PARAGRAPH DISCUSSING MEN’S ATTITUDES: “IT IS POSSIBLE THAT THE LACK OF A STRONG ASSOCIATION IN OUR SAMPLE IS DUE TO SOCIAL DESIRABILITY BIAS IN MEN’S REPORTING OF ATTITUDES WHICH MIGHT BE EXACERBATED BY THEIR KNOWLEDGE OF THEIR PARTNER’S INVOLVEMENT IN AN INTERVENTION ON HEALTHY RELATIONSHIPS.”

The discussion does not provide explanation about what could be happening with women who did not disclose IPV but still were unwilling for their male partners to participate. These may be an important group to understand. 

TA: THANKS FOR THIS USEFUL COMMENT. WE HAD ALREADY INCLUDED DISCUSSION OF NON-IPV RELATED FACTORS ASSOCIATED WITH CONSENT/LACK OF CONSENT – FOR EXAMPLE LOWER CONSENT AMONG WOMEN IN SHORTER TERM RELATIONSHIPS. IN THE SECOND PARAGRAPH OF THE DISCUSSION, WE HAVE NOW INCLUDED MORE EXPLICIT DISCUSSION OF WHY THIS MIGHT BE: “IT IS POSSIBLE THAT THOSE IN NEWER RELATIONSHIPS FEEL LESS COMFORTABLE BEING ASSOCIATED WITH DEMANDS ON THEIR PARTNER’S TIME. WOMEN IN RELATIONSHIPS WITH POORER COMMUNICATION MIGHT ALSO BE LESS LIKELY TO HAVE TOLD THEIR PARTNER ABOUT THEIR OWN PARTICIPATION IN MAISHA OR BE CONCERNED THAT HE WOULD REACT NEGATIVELY TO HER SHARING HIS DETAILS WITH THE STUDY TEAM.”

Limitations

A missed opportunity is that researchers did not explore women’s motivations for their un/willingness to let their male partners participate in the study. It is important to cite this as a limitation. However, the speculated explanations based on data distribution of women who consented is well argued. 

TA: SEE ABOVE RESPONSE.

IN ADDITION TO THE POINTS RAISED ABOVE, WE HAD ALSO ALREADY INCLUDED DISCUSSION OF HOW:

• WOMEN’S IPV EXPERIENCE AND CONCERNS FOR THEIR OWN SAFETY MAY MOTIVATE THEM TO WITHHOLD CONSENT IN THE CONTROL GROUP

• WOMEN’S POSITIVE EXPERIENCES OF PARTICIPATING IN THE INTERVENTION MAY PREDISPOSE THEM TO CONSENT TO THEIR PARTNER BEING APPROACHED, PERHAPS EVEN MORE SO IN CASES WHERE THE PARTNER IS VIOLENT AND THEY FEEL CONTACT WITH THE TRIAL TEAM MIGHT BE BENEFICIAL. 

**

Editor’s comments

http://www.journals.plos.org/plosone/s/file?id=wjVg/PLOSOne_formatting_sample_main_body.pdf andhttp://www.journals.plos.org/plosone/s/file?id=ba62/PLOSOne_formatting_sample_title_authors_affiliations.pdf

TA: WE HAVE RENAMED THE SUPPLEMENTARY FILES TO COMPLY WITH PLOS ONE REQUIREMENTS. THEY NOW INCLUDE THE PREFIX S1, S2, ETC. 

TA: WE HAVE INCLUDED THE QUESTIONNAIRES IN ENGLISH AND SWAHILI AS SUPPLEMENTARY FILES 1-4. 

3. Thank you for stating in your financial disclosure: 

"The MAISHA study was funded by an anonymous donor and supported by the STRIVE Research Consortium, which is funded by UK Aid (Grant number PO5244 held by SL) from the UK government Department for International Development (https://www.ukaiddirect.org/) . The views expressed in the paper do not necessarily reflect the Department’s official policies. The funders had no role in study design, data collection and analysis, decision to publish, or preparation of the manuscript. "

PLOS ONE requires you to include in your manuscript further information about the funder so that any relevant competing interests can be assessed. Please respond to the following questions:cc

a) Please state whether any of the research costs or authors' salaries were funded, in whole or in part, by a tobacco company (our policy on tobacco funding is athttp://journals.plos.org/plosone/s/disclosure-of-funding-sources) 

TA: NONE OF THE RESEARCH COSTS OR AUTHORS’ SALARIES WERE FUNDED, IN WHOLE OR IN PART, BY A TOBACCO COMPANY. 

b) Please state whether the donor has any competing interests in relation to this work (see http://journals.plos.org/plosone/s/competing-interests) .

TA: THE DONORS HAVE NO COMPETING INTERESTS IN RELATION TO THIS WORK. 

c) Please state whether the identity of the donor might be considered relevant to editors or reviewers’ assessment of the validity of the work.

TA: THE DONORS’ IDENTITIES WOULD NOT BE CONSIDERED RELEVANT TO EDITORS OR REVIEWERS’ ASSESSMENT OF THE VALIDITY OF THE WORK. 

d) If the donors have no perceived or actual competing interests, please state: “The authors are not aware of any competing interests”.

TA: THE AUTHORS ARE NOT AWARE OF ANY COMPETING INTERESTS. 

This information should be included in your cover letter. We will amend your financial disclosure and competing interests on your behalf.

TA: WE DO NOT WISH TO CHANGE OUR DATA AVAILABILITY STATEMENT. WE ARE IN THE PROCESS OF PREPARING THE DATASETS FOR THE LSHTM DATA REPOSITORY. DUE TO THE CURRENT LOCKDOWN SITUATION AND WORKING FROM HOME ARRANGEMENTS, THIS PROCESS MAY TAKE SLIGHTLY LONGER THAN ANTICIPATED. UNTIL THE URL IS AVAILABLE, WE WOULD BE HAPPY TO HANDLE DATA REQUESTS SUBMITTED DIRECTLY TO THE CORRESPONDING AUTHOR.

---

## [Decision Letter · Decision Letter 1]

3 Aug 2020

PONE-D-20-05439R1

Couples data from north-western Tanzania: Insights from a survey of male partners of women enrolled in the MAISHA cluster randomized trial of an intimate partner violence prevention intervention

PLOS ONE

Dear Dr. Abramsky,

Thank you for submitting your manuscript to PLOS ONE. After careful consideration, we feel that it has merit but does not fully meet PLOS ONE’s publication criteria as it currently stands. Therefore, we invite you to submit a revised version of the manuscript that addresses the points raised during the review process.

We look forward to receiving your revised manuscript.

Kind regards,

Thach Duc Tran, M.Sc., Ph.D.

Academic Editor

PLOS ONE

Reviewers' comments:

Reviewer's Responses to Questions

**Comments to the Author**

1. If the authors have adequately addressed your comments raised in a previous round of review and you feel that this manuscript is now acceptable for publication, you may indicate that here to bypass the “Comments to the Author” section, enter your conflict of interest statement in the “Confidential to Editor” section, and submit your "Accept" recommendation.

Reviewer #2: All comments have been addressed

2. Is the manuscript technically sound, and do the data support the conclusions?

Reviewer #2: Yes

3. Has the statistical analysis been performed appropriately and rigorously? 

Reviewer #2: Yes

4. Have the authors made all data underlying the findings in their manuscript fully available?

Reviewer #2: No

5. Is the manuscript presented in an intelligible fashion and written in standard English?

Reviewer #2: Yes

6. Review Comments to the Author

Reviewer #2: 

Thank you for the opportunity to review the paper: Couples data from north-western Tanzania: Insights from a survey of male partners of women enrolled in the MAISHA cluster randomized trial of an intimate partner violence prevention intervention. Please see my comments below

Title: The title now reflects better on the contents of the paper

Interpretation of results:

The reflection of the authors on the usefulness of the p-value in determining significance is reasonable. The changes they made based on directions of the associations are acceptable. However, it will be desirable to have some of this explanation – i.e consideration of direction of associations over significance levels provide in the statistical analysis section of the paper.

Discussion:

• Authors have appropriately addressed the comment around potential of social desirability bias influencing men's responses.

• I am satisfied with the explicit discussion of why women in newer relationships may have given less consent- i.e they may have been less comfortable, have poorer communication such that their partner was not aware of their participation, or unsure of the demands on their partner's and reactions.

• The authors have added satisfactory explanation about how they ensured minimal risk to participants whose partners were abusive.

7. PLOS authors have the option to publish the peer review history of their article (what does this mean?). If published, this will include your full peer review and any attached files.

Reviewer #2: No

---

## [Author Response · Author response to Decision Letter 1]

11 Sep 2020

Dear Editor, 

Thank you for considering our manuscript: "Couples data from north-western Tanzania: Insights from a survey of male partners of women enrolled in the MAISHA cluster randomized trial of an intimate partner violence prevention intervention." We are pleased that the reviewers are satisfied with the changes we made to the original manuscript. The two outstanding comments are addressed below:

1) We recently submitted an Email via editorial manager to advise of a change to our data availability statement. The data have now been submitted to the London School of Hygiene and Tropical Medicine (LSHTM) data archive, Data Compass( DOI: https://doi.org/10.17037/DATA.00001773 ). However, the LSHTM Research Data Management team have raised ethical/data protection concerns regarding open access to this data. This research was performed prior to data sharing requirements being established and participant consent for sharing was not obtained. For this reason, and due to the sensitivity of the research topic, we have been advised that it will not be possible to make the datasets openly available. Instead, access to an anonymised subset will be provided for use in ethically approved research, on condition those requesting access comply with ethical conditions associated with the study. Researchers are invited to apply for data access, outlining the analysis they wish to perform and the data variables requested. If the proposed use is approved, they will be asked to sign a Data Transfer Agreement that requires them to keep data confidential.

2) In relation to reviewer 2’s comment that the statistical analysis section should include explanation of our consideration of the direction as well as p-values of associations, we have added the following paragraph to the statistical analysis section of the manuscript: “In recognition of low study power in relation to some of the associations under study (the CRT sample size having been determined around power to detect intervention impacts on primary outcomes), interpretation of results will involve consideration of the magnitude and direction of observed associations in addition to p-values and 95% confidence intervals.”

Thank you again for considering our paper. Please let us know if you require any more information regarding data availability. We have submitted a revised manuscript with tracked changes, as well as an unmarked copy. We look forward to hearing from you. 

Yours Sincerely,

Tanya Abramsky

---

## [Editor Report · Decision Letter 2]

21 Sep 2020

Couples data from north-western Tanzania: Insights from a survey of male partners of women enrolled in the MAISHA cluster randomized trial of an intimate partner violence prevention intervention

PONE-D-20-05439R2

Dear Dr. Abramsky,

We’re pleased to inform you that your manuscript has been judged scientifically suitable for publication and will be formally accepted for publication once it meets all outstanding technical requirements.

Kind regards,

Thach Duc Tran, M.Sc., Ph.D.

Academic Editor

PLOS ONE
---

## [Editor Report · Acceptance letter]

24 Sep 2020

PONE-D-20-05439R2 

Couples data from north-western Tanzania: Insights from a survey of male partners of women enrolled in the MAISHA cluster randomized trial of an intimate partner violence prevention intervention 

Dear Dr. Abramsky:

I'm pleased to inform you that your manuscript has been deemed suitable for publication in PLOS ONE. Congratulations! Your manuscript is now with our production department. 

Kind regards, 

on behalf of

Dr. Thach Duc Tran 

Academic Editor

PLOS ONE